# SCAN: Self-Denoising Monte Carlo Annotation for Robust Process Reward Learning

**Yuyang Ding**[1], **Xinyu Shi**[1], **Juntao Li**[1,*], **Xiaobo Liang**[1], **Zhaopeng Tu**[2], **Min Zhang**[1]

[1]Soochow University          [2]Tencent

{yyding23,xyshi02}@stu.suda.edu.cn
{ljt,xbliang,minzhang}@suda.edu.cn   zptu@tencent.com

Project Page: https://scan-prm.github.io

## Abstract

Process reward models (PRMs) offer fine-grained, step-level evaluations that facilitate deeper reasoning processes in large language models (LLMs), proving effective in complex tasks like mathematical reasoning. However, developing PRMs is challenging due to the high cost and limited scalability of human-annotated data. Synthetic data from Monte Carlo (MC) estimation is a promising alternative but suffers from a high noise ratio, which can cause overfitting and hinder large-scale training. In this work, we conduct a preliminary study on the noise distribution in synthetic data from MC estimation, identifying that annotation models tend to both underestimate and overestimate step correctness due to limitations in their annotation capabilities. Building on these insights, we propose **S**elf-Denoising Monte **C**arlo **An**notation (SCAN), an efficient data synthesis and noise-tolerant learning framework. Our key findings indicate that: (1) Even lightweight models (e.g., 1.5B parameters) can produce high-quality annotations through a self-denoising strategy, enabling PRMs to achieve superior performance with only 6% the inference cost required by vanilla MC estimation. (2) With our robust learning strategy, PRMs can effectively learn from this weak supervision, achieving a 39.2 F1 score improvement (from 19.9 to 59.1) in ProcessBench. Despite using only a compact synthetic dataset, our models surpass strong baselines, including those trained on large-scale human-annotated datasets such as PRM800K. Furthermore, performance continues to improve as we scale up the synthetic data, highlighting the potential of SCAN for scalable, cost-efficient, and robust PRM training.

## 1 Introduction

The recent advent of large language models (LLMs) such as OpenAI o1 [1, 13] and DeepSeek R1 [9] has sparked significant interest in scaling test-time compute to encourage slower and deeper reasoning processes. In this context, process reward models [17, 35] have emerged as a promising approach, offering fine-grained, step-level evaluations that facilitate iterative self-refinement [29] and exploration of solution spaces [18]. This proves to be particularly effective in tackling complex problems like mathematical reasoning tasks.

However, a critical challenge for developing process reward models (PRMs) lies in data annotation. While human-annotated methods [17] can produce high-quality data that effectively guides PRM training, they come at a high cost. To address this, many works [35, 18, 26] have explored synthetic data generation via Monte Carlo estimation. However, synthetic data still falls short of matching the quality of human-annotated data, as explored by Zheng et al. [44]. The primary challenge stems

---

*Corresponding author

39th Conference on Neural Information Processing Systems (NeurIPS 2025).

from the high noise ratio inherent in Monte Carlo-generated data, as models tend to quickly overfit noisy samples, hindering effective training at larger data scales. Recent studies [30, 28, 43] have demonstrated that introducing stronger supervision from large-scale critic models (e.g., Qwen-72B) is an effective strategy for mitigating noise by retaining only consensus samples agreed upon by both the critic model and Monte Carlo estimation [5, 14]. However, how noise is distributed and how to train PRMs robustly in the presence of such noise are still underexplored. **In this paper, we investigate the full potential of denoising in MC estimation itself and robust learning in PRMs, without relying on any external stronger supervision.** *To the best of our knowledge, we are the first to systematically explore Process Reward Learning from the perspective of noise distribution and robust learning.*

We begin with a preliminary study to investigate the noise distribution present in synthetic data generated through Monte Carlo estimation. Our findings indicate that this noise arises mainly from two factors: the annotation model tends to under-estimate and over-estimate step correctness, primarily due to inherent limitations in its annotation capabilities. To address this issue, we introduce a self-confidence metric designed to assess the reliability of model-generated annotations. Guided by these insights, we propose strategies to mitigate two distinct types of noise: noisy positive samples and inaccurate negative samples. We develop a selective sampling approach to reduce overall sample noise and design a model-wise self-denoising loss for robust learning. By leveraging the self-confidence metric, we systematically reduce the annotation model's bias and enhance the overall data quality. Furthermore, to improve data synthesis efficiency, we selectively apply Monte Carlo annotation to the most informative samples, optimizing both accuracy and computational resources.

Using weak supervision from a lightweight annotation model, Qwen2.5-Math-1.5B-Instruct [39], we construct a synthetic dataset consisting of 101K samples. We evaluate the trained PRM from two perspectives: test-time scaling (best-of-8 evaluation) and step-wise error detection (ProcessBench [44]). The results demonstrate that our model consistently outperforms strong existing baselines, achieving performance comparable to that of the human-annotated dataset PRM800K. Further scaling with an additional 97K synthetic samples generated by Llama3.2-3B-Instruct and Qwen2.5-Math-7B-Instruct leads to our PRM surpassing the performance of the human-annotated PRM800K on both evaluation benchmarks. As the dataset size continues to grow, we observe further scaling potential.

## 2 Preliminary: Unveiling the Noise Distribution in Monte Carlo Annotaion

### 2.1 Problem Definition

**Monte Carlo Estimation**  In the context of process annotation, Monte Carlo estimation [35] is proposed as an automated approach for evaluating the correctness of each step. Formally, given a question $q$ and a corresponding response $\mathbf{x}$ containing $n$ steps, i.e., $\mathbf{x} = [x_1, x_2, \ldots, x_n]$, the correctness score of the $t^{th}$ step, denoted as $c_t$, can be estimated by a completer model through multiple rollouts or simulations. Specifically, the completer model is prompted to sample $k$ completions based on the question $q$ and prefix steps $\mathbf{x}_{\leq t} = [x_1, \ldots, x_t]$ until a terminate state is reached.

$$c_t = \mathbb{E}_{r \sim P_\theta(\cdot|q, \mathbf{x}_{\leq t})}[\mathcal{J}(r, a^*)], \quad \hat{c}_t = \frac{1}{k}\sum_{i=1}^{k} \mathcal{J}(r^{(i)}, a^*), \quad (1)$$

where $\mathcal{J}(r, a^*)$ is an indicator function that equals 1 if the sampled response $r$ matches the ground truth answer $a^*$, and 0 otherwise.

**PRM Training**  After obtaining the correctness score $c_t$ for each step, we can train the PRM using binary classification loss:

$$\mathcal{L}_{\text{BCE}}(\theta) = -\mathbb{E}_{(x_{\leq t}, y_t) \sim D_{\text{train}}}[y_t \log(P_\theta(y_t|q, \mathbf{x}_{\leq t})) + (1 - y_t)\log(1 - P_\theta(y_t|q, \mathbf{x}_{\leq t}))], \quad (2)$$

where $y_t$ represents the correctness label of for the $t^{th}$ step. We use hard labels to annotate $y_t$, defined as $y_t = \mathbf{1}[c_t > 0]$, assigning $y_t = 1$ for any positive correctness score and $y_t = 0$ otherwise. We consider a step correct if at least one rollout leads to the correct final answer.

**Noise issue in MC estimation**  A significant effectiveness gap exists between synthetic data and human-annotated data [44] for training PRMs. This discrepancy can be attributed to the noise label

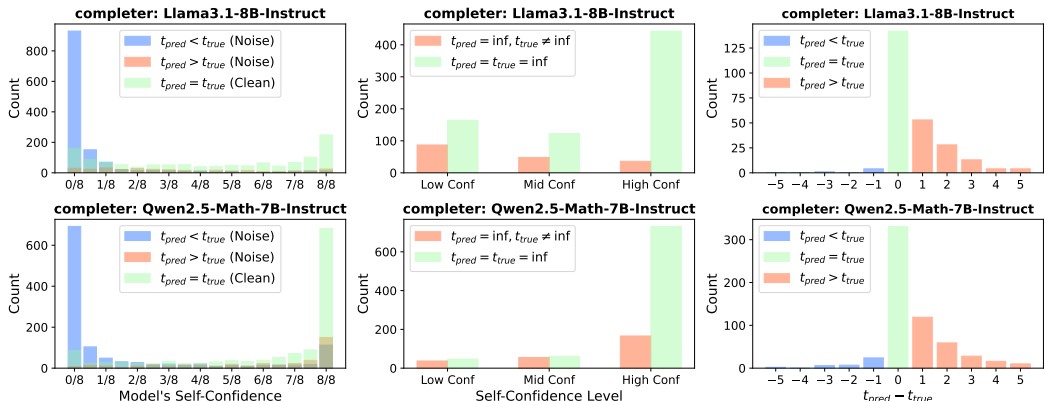

Figure 1: Noise distribution analysis of Llama3.1-8B-Instruct and Qwen2.5-Math-7B-Instruct. **Left:** Overall distribution of noise samples across varying self-confidence levels. **Middle:** Noise distribution in predicted positive samples where $t_{pred} = \inf$. **Right:** Distance distribution between $t_{pred}$ and $t_{true}$ for inaccurate negative samples. Additional results of more models can be found in Figure 4.

issue inherent in Monte Carlo estimation [18, 42], where the correctness of each reasoning step is often misestimated, primarily due to the limitations of the completer model. More specifically, the correctness of the current step is annotated by the completer model based on whether the completions lead to the correct final answer. As a result, the PRM trained on this data only estimates the potential of the "completer model" to reach the correct final answer from the current step. This differs significantly from the correctness of the current step, as the completer model is not perfect and can make mistakes. This is why the noise phenomenon exists in Monte Carlo estimation.

## 2.2 Noise Distribution in MC estimation

Given a question $q$ and a corresponding response $\mathbf{x} = [x_1, x_2, \ldots, x_n]$, we focus only on the first error step during the training [17] and evaluation [44] of the PRM. We denote the first error location predicted by the completer through Monte Carlo estimation as $t_{pred}$, while the ground truth error location is $t_{true}$. For cases where the response is entirely correct, we use the label "inf" (i.e., $+\infty$) to indicate the absence of any errors.

Then we can categorize the mistakes made by the completer model into the following two types (fully correct samples are also taken into account):

- **Under-Estimation** ($t_{pred} < t_{true}$): This likely occurs because the completer struggles with complex or nuanced reasoning. Even when provided with correct prefix steps, it may fail to generate correct rollouts, leading to early error detection.
- **Over-Estimation** ($t_{pred} > t_{true}$): This happens when the completer initially corrects an error, producing a correct rollout. However, as subsequent errors accumulate, the model cannot fully address them, causing delayed detection of the true error location.

To quantify this relationship between the noise distribution and the completer's problem-solving capability, we propose a metric called **self-confidence**, which measures the confidence level of the completer model on the annotated question:

$$SC_\theta(q) = \mathbb{E}_{r \sim P_\theta(\cdot|q)} \mathcal{J}(r, a^*), \tag{3}$$

where $\mathcal{J}(r, a^*)$ is an indicator function that evaluates whether the response $r$ matches the ground truth answer $a^*$. In practice, $SC_\theta(q)$ is estimated by computing the empirical mean over multiple randomly sampled responses given question $q$.

**Experimental Settings** We select four representative open-source models, i.e., Llama3.1-8B-Instruct, Llama3.2-3B-Instruct [6], Qwen2.5-Math-1.5B-Instruct, and Qwen2.5-Math-7B-Instruct [39], as the completer models. For the dataset, we use ProcessBench [44], which contains

3,400 human-annotated process data points spanning multiple difficulty levels. To compute self-confidence, we sample 16 completions for each question. For evaluating step correctness, we perform 8 rollouts per step to determine its correctness.

**Key Observations**   The left of figure 1 illustrates the overall noise distribution of different models across different self-confidence values.[2] We mainly have the following observations:

- **Observation 1:** Noisy cases where $t_{pred} < t_{true}$ are predominantly concentrated in low self-confidence regions, supporting the under-estimation hypothesis discussed before.
- **Observation 2:** The noise distribution of $t_{pred} > t_{true}$ varies across models. For the more capable Qwen model, its stronger error-correction ability results in noise being concentrated in high self-confidence regions. In contrast, noise is more evenly distributed for the Llama model.
- **Observation 3:** Clean samples ($t_{pred} = t_{true}$) are primarily located in high self-confidence regions. Overall, high-confidence regions exhibit a lower proportion of noise.

### 2.3   Detailed Analysis of Noise Distribution

At a finer granularity, we classify noise into two distinct categories: **Noisy Positive Samples** and **Inaccurate Negative Samples**.

**Noisy Positive Samples**   Samples observed as fully correct positives with $t_{pred} = \inf$, but actually contain errors, i.e., $t_{true} \neq \inf$. These cases indicate that the model fails to detect existing errors, leading to noisy positives in its predictions. For clarity, we categorize self-confidence into three levels: low confidence as [0, 0.25], medium confidence as (0.25, 0.75), and high confidence as [0.75, 1]. These thresholds are empirically determined through the overall noise distribution. The middle of figure 1 illustrates the distribution of noise positive samples, from which we can conclude that:

- **Observation 4:** For predicted positive samples ($t_{pred} = \inf$), the noise positive ratio is significantly lower in high self-confidence samples, making them more suitable for training.

**Inaccurate Negative Samples**   Samples observed as containing errors ($t_{pred} \neq \inf$), but the predicted error location is incorrect, i.e., $t_{pred} \neq t_{true}$. This reflects the model's inability to precisely identify the true error location, even when it detects the presence of errors. From **Observations 1, 3, 4**, we can conclude that the completer model makes fewer mistakes in high self-confidence samples. Therefore, we focus on the high-confidence subset to investigate how noise is distributed in inaccurate negative samples. We visualize the relationship between $t_{pred}$ and $t_{true}$ in the right of figure 1, from which we can conclude:

- **Observation 5:** In most cases, the model can roughly predict the error location but often lacks precision. Furthermore, the model tends to overestimate the error location, i.e., $t_{pred} > t_{true}$, and the number of noisy samples decreases as the deviation increases. This also supports the over-estimation hypothesis discussed before.

## 3   Methodology

Building on the observations of noise distribution discussed in section 2, we propose targeted approaches to address key challenges in Monte Carlo annotation. Specifically, our method contains two modules: (1) an efficient data synthesis framework to reduce substantial inference costs, and (2) robust training methods to mitigate the high noise ratio in synthetic data and enable robust learning with noisy labels. Figure 2 illustrates the overall workflow of our proposed method.

### 3.1   Efficient Data Synthesis Framework

**Generate Responses**   The synthesis process begins with a dataset of questions and their corresponding golden answers, denoted as $D = \{(q_i, a_i)\}_{i=1}^{M}$. For each question $q_i$, a generator model

---

[2]We include fully correct samples, assigning them an error location label of $+\infty$ for convenience.

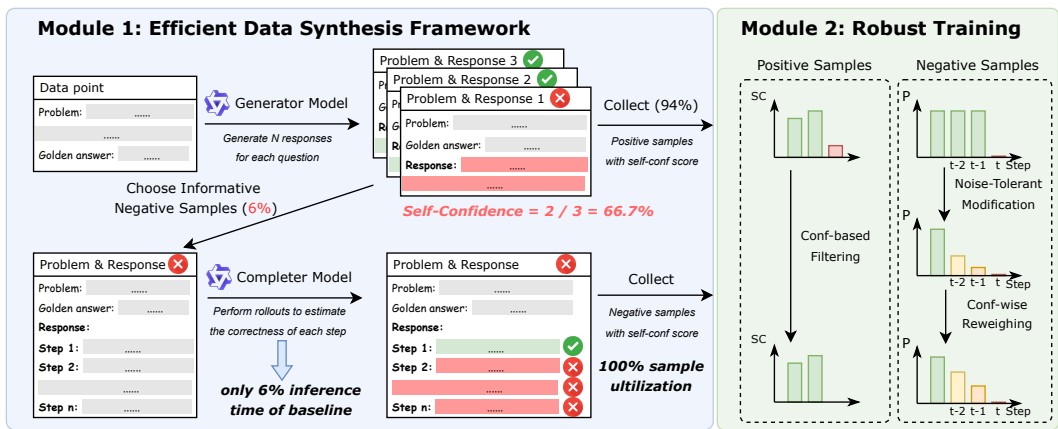

Figure 2: Overview of our data synthesis and robust training framework.

parameterized by $\pi_1$ produces $N$ responses, denoted as $\{r_{i,1}, r_{i,2}, \ldots, r_{i,N}\}$. The confidence score of the generator in $q_i$ is then computed as:

$$SC_{\pi_1}(q_i) = \frac{1}{N} \sum_{j=1}^{N} \mathcal{J}(r_i^{(j)}, a_i), \quad \text{where } r_i \sim P_\theta(\cdot \mid q_i) \tag{4}$$

where $\mathcal{J}(r_i^{(j)}, a^i)$ evaluates the correctness of the generated response. Next, we collect the negative samples with $t_{pred} \neq \text{inf}$ from these responses for further step-wise annotation. In this process, we do not select the annotated correct (positive) samples for subsequent annotation. Although these positive samples may contain false positives that could potentially be filtered out through detailed step-wise correctness checks, we find that the annotation cost for this process is prohibitively high. For instance, performing 8 rollouts for each step in a 10-step response requires $8 \times 10 = 80$ rollouts per sample. Moreover, the positive samples, particularly those in high self-confidence regions, contain minimal noise (from **Observation 4**). Therefore, we directly use these high-confidence samples as positive examples for training without further annotation. By applying Monte Carlo estimation exclusively to negative samples, we ensure 100% sample utilization—every sample annotated through Monte Carlo estimation is included in the final training dataset.

**Step-wise correctness annotation** Negative samples with high self-confidence scores, $SC_{\pi_1}(q_i) > \epsilon$, are selected for step-wise correctness annotation using a completer model parameterized by $\pi_2$. We employ vLLM [15] and implement distributed inference via Ray [22] to accelerate the annotation process. To support the subsequent robust learning process, we also need to collect the self-confidence scores $SC_{\pi_2}(q_i)$ during annotation. For convenience, we use the same model as the generator and completer models, i.e., $\pi_1 = \pi_2 = \pi$, allowing us to reuse the self-confidence scores and further enhance data generation efficiency.

### 3.2 Robust Learning with Noisy Labels

The final annotated dataset, denoted as $D_{final}$, consists of tuples $(q, \mathbf{x}, \mathbf{c}, SC_\pi(q))$, where $q$ is the question, $\mathbf{x} = [x_1, x_2, \ldots, x_n]$ represents the $n$-step responses, and $\mathbf{c} = [c_1, c_2, \ldots, c_n]$ contains the corresponding correctness scores annotated via Monte Carlo (MC) estimation. The term $SC_\pi(q)$ denotes the self-confidence score of the completer model $\pi$ for question $q$. We then train PRMs with the reweighted step label:

$$\mathcal{L}_{\text{SCAN}}(\theta) = -\mathbb{E}_{(x_{\leq t}, y_t) \sim D_{\text{final}}}[y_t \log(P_\theta(y_t|q, \mathbf{x}_{\leq t})) + (1 - y_t) \log(1 - P_\theta(y_t|q, \mathbf{x}_{\leq t}))]$$

$$\hat{y}_t = \begin{cases} \min(c_t/SC_\pi(q), 1), & \text{if } t^e_{pred} - t \leq d \\ \mathbb{I}(c_t > 0), & \text{Otherwise} \end{cases}, \quad \text{where } c_t = P_\pi(y_t = \text{correct}|q, \mathbf{x}_{\leq t}),$$
$$\tag{5}$$

where $t^e_{pred}$ denotes the first error location with $c_t = 0$. Compared to the traditional BCE loss, our modifications focus on two main aspects: noise-tolerant labeling and confidence-wise reweighting.

**Noise-tolerant Labeling**    The completer model tends to overestimate the correctness of the current step due to its strong self-correction capability. As errors continue to accumulate, the model eventually makes mistakes, leading to $t_{pred} > t_{true}$, with a high probability of similar errors occurring at nearby positions (**Observation 5**). To enable more robust learning with these noisy labels, we propose a noise-tolerant labeling strategy that applies soft labels to steps preceding the error, within a tolerance distance $d$. We discuss the choice of $d$ in section 4.4. Through experiments, we demonstrate that the strategy allows the PRM to learn more effectively from noisy labels without overfitting.

**Confidence-wise Reweighting**    After applying the denoising process, a critical issue remains: the annotated labels are still heavily influenced by the capability of the completer model, i.e.,

$$c_t^* = P(y_i = \text{correct}|\pi_{\text{gold}}, q, \mathbf{x}_{\leq t}), \quad \text{while } c_i = P(y_i = \text{correct}|\pi_\theta, q, \mathbf{x}_{\leq t}). \tag{6}$$

Here, $c_i^*$ represents the true correctness probability assigned by an assumed golden annotator $\pi_{\text{gold}}$, while $c_i$ is the estimated probability derived from the completer model $\pi_\theta$. Since we can only estimate $c_i$ through rollouts performed by the imperfect completer $\pi_\theta$, the true label $c_i^*$ remains unobservable. To mitigate the model-dependent bias, we introduce a correction factor $\delta_i = c_i^*/c_i$. We leverage the self-confidence score $SC_\pi(q)$ to approximate and reduce this model-induced bias. The rationale behind this approach is as follows: consider two models annotating the same sample—a strong model with $SC_{\text{strong}}(q)$ and a weak model with $SC_{\text{weak}}(q)$. Naturally, the stronger model will yield a higher correctness score $c_i$. However, we aim for the final corrected scores to be consistent across models, regardless of their inherent strengths. Thus, we adjust the estimated correctness score using the self-confidence score as follows:

$$\hat{c}_i^* = \min(c_i/SC_\pi(q), 1). \tag{7}$$

This adjustment helps to normalize the influence of model capability on the annotated labels, leading to more reliable and unbiased training data. This reweighting procedure is particularly effective when integrating annotations from multiple completer models, as demonstrated by the experimental results in section 4.3.

# 4    Experiment

## 4.1    Training Dataset Construction

**SCAN-Base (101K)**    We use 7,500 questions with golden answers from the MATH training set. For each question, we generate $k$ responses and perform eight rollouts per step to annotate process correctness via MC estimation. We set tolerance distance $d$ to 2, with discussion in section 4.4. SCAN-Base is generated by Qwen2.5-Math-1.5B-Instruct [39] as both the generator and completer. We experiment with varying numbers of responses from $k \in \{64, 128\}$ and found that a larger number of responses provides a more accurate estimation of the model's self-confidence, which is crucial for denoising and robust learning processes, as elaborated in Appendix B.4. Additionally, we explore other data sources, including GSM8K [2] and Numina-Math [16], and discover that the MATH dataset offers a more suitable level of difficulty and higher data quality for training (Appendix B.4).

**SCAN-Pro (197K)**    We further incorporate Qwen2.5-Math-7B-Instruct [39] and Llama3.2-3B-Instruct [6] for Monte Carlo estimation. With these two models, we generate an additional 97K data points. Integrated with SCAN-Base, we construct a mixed dataset containing 197K samples. More dataset details are shown in Appendix A.1. We observe that (1) combining these two datasets increases the diversity of the data, thereby boosting model performance, and (2) the model-wise reweighted loss design mitigates any data inconsistency arising from the capability gap between the annotation models, with experiment results in section 4.4.

## 4.2    Experimental Setup

We evaluate the effectiveness of the Process Reward Model (PRM) from two key perspectives:

**Best-of-N (BoN) Evaluation**    In this evaluation, the PRM functions as a verifier to select the best response from multiple candidate answers generated by a policy model. Specifically, the PRM assigns scores to each step within a response and then aggregates these step-wise scores into an overall

Table 1: Best-of-8 evaluation results of the policy model Qwen2.5-Math-7B-Instruct. Results of policy model Llama3.1-8B-Instruct can be found in Appendix B.3. To reduce potential errors, we re-evaluated all these models based on the same set of responses.

| Model | Training Samples | Annotation Method | GSM8K | MATH | College Math | Olympiad Bench | Avg. |
|---|---|---|---|---|---|---|---|
| Greedy | | | 95.6 | 83.6 | 46.9 | 40.6 | 66.7 |
| Majority Vote@8 | | | 96.9 | 87.3 | 47.4 | 43.0 | 68.7 |
| **Pass@8 (Upper Bound)** | | | 98.0 | 92.0 | 52.3 | 60.4 | 75.7 |
| UniversalPRM-7B | 690K | MC, KD | 96.8 | 86.9 | 47.6 | 48.0 | 69.8 |
| Qwen2.5-Math-PRM-7B | 1500K | MC, KD | 96.8 | 88.1 | 47.7 | 47.6 | 70.1 |
| RLHFlow-PRM-Mistral-8B | 273K | MC-Only | 97.1 | 87.3 | 47.3 | 43.0 | 68.7 |
| RLHFlow-PRM-DeepSeek-8B | 253K | MC-Only | 96.8 | 87.3 | 47.9 | 43.9 | 69.0 |
| EurusPRM-Stage1 | 463K | Implicit | 96.9 | 86.0 | 47.4 | 42.8 | 68.3 |
| EurusPRM-Stage2 | 693K | Implicit | 97.0 | 86.7 | 47.9 | 45.3 | 69.2 |
| Math-PSA-7B | 1395K | MC-Only | 96.4 | 86.0 | 47.7 | 45.9 | 69.0 |
| Qwen2.5-Math-7B-Math-Shep | 445K | MC-Only | 96.9 | 86.8 | 47.6 | 42.3 | 68.3 |
| Skywork-PRM-Qwen2.5-7B | - | - | 97.0 | 87.9 | 47.8 | 44.6 | 69.3 |
| Qwen2.5-Math-7B-PRM800K | 264K | Human | 97.0 | 87.6 | 47.7 | 45.0 | 69.3 |
| Qwen2.5-Math-7B-SCAN-Base | 101K | MC-Only | 97.1 | 86.9 | 47.8 | 44.4 | 69.1 |
| Qwen2.5-Math-7B-SCAN-Pro | 197K | MC-Only | **97.1** | 87.3 | **48.1** | **47.7** | **70.1** |

Table 2: Evaluation Results on ProcessBench. MC denotes Monte Carlo estimation, while KD represents knowledge distillation from more capable critic models (with 32B or larger parameters).

| Model | GSM8K | | | MATH | | | Olympiad Bench | | | Omni Math | | | Avg. |
|---|---|---|---|---|---|---|---|---|---|---|---|---|---|
| | error | correct | **F1** | error | correct | **F1** | error | correct | **F1** | error | correct | **F1** | **F1** |
| **Criric Models (LLM-as-a-judge)** | | | | | | | | | | | | | |
| GPT-4o-0806 | 70.0 | 91.2 | 79.2 | 54.4 | 76.6 | 63.6 | 45.8 | 58.4 | 51.4 | 45.2 | 65.6 | 53.5 | 61.9 |
| Qwen2.5-Math-7B-Instruct | 15.5 | 100.0 | 26.8 | 14.8 | 96.8 | 25.7 | 7.7 | 91.7 | 14.2 | 6.9 | 88.0 | 12.7 | 19.9 |
| Llama-3.3-70B-Instruct | 72.5 | 96.9 | 82.9 | 43.3 | 83.2 | 59.4 | 31.0 | 94.1 | 46.7 | 28.2 | 90.5 | 43.0 | 58.0 |
| Qwen2.5-72B-Instruct | 62.8 | 96.9 | 76.2 | 46.3 | 93.1 | 61.8 | 38.7 | 92.6 | 54.6 | 36.6 | 90.9 | 52.2 | 61.2 |
| QwQ-32B-Preview | 81.6 | 95.3 | 88.0 | 78.1 | 79.3 | 78.7 | 61.4 | 54.6 | 57.8 | 55.7 | 68.0 | 61.3 | 71.5 |
| **Process Reward Models (MC + KD)** | | | | | | | | | | | | | |
| UniversalPRM-7B | - | - | 85.8 | - | - | 77.7 | - | - | 67.6 | - | - | 66.4 | 74.3 |
| Qwen2.5-Math-PRM-7B | 72.0 | 96.4 | 82.4 | 68.0 | 90.4 | 77.6 | 55.7 | 85.5 | 67.5 | 55.2 | 83.0 | 66.3 | 73.5 |
| **7-8B Process Reward Models (w/o KD)** | | | | | | | | | | | | | |
| RLHFlow-PRM-Mistral-8B | 33.8 | 99.0 | 50.4 | 21.7 | 72.2 | 33.4 | 8.2 | 43.1 | 13.8 | 9.6 | 45.2 | 15.8 | 28.4 |
| RLHFlow-PRM-Deepseek-8B | 24.2 | 98.4 | 38.8 | 21.4 | 80.0 | 33.8 | 10.1 | 51.0 | 16.9 | 10.9 | 51.9 | 16.9 | 26.6 |
| EurusPRM-Stage1 | 46.9 | 42.0 | 44.3 | 33.3 | 38.2 | 35.6 | 23.9 | 19.8 | 21.7 | 21.9 | 24.5 | 23.1 | 31.2 |
| EurusPRM-Stage2 | 51.2 | 44.0 | 47.3 | 36.4 | 35.0 | 35.7 | 25.7 | 18.0 | 21.2 | 23.1 | 19.1 | 20.9 | 31.3 |
| Qwen2.5-Math-7B-Math-Shep | 46.4 | 95.9 | 62.5 | 18.9 | 96.6 | 31.6 | 7.4 | 93.8 | 13.7 | 4.0 | 95.0 | 7.7 | 28.9 |
| Skywork-PRM-Qwen2.5-7B | 61.8 | 82.9 | 70.8 | 43.8 | 62.2 | 53.6 | 17.9 | 31.9 | 22.9 | 14.0 | 41.9 | 21.0 | 42.1 |
| Qwen2.5-Math-7B-PRM800K | 53.1 | 95.3 | 68.2 | 48.0 | 90.1 | 62.6 | 35.7 | 87.3 | **50.7** | 29.8 | 86.3 | 44.3 | 56.5 |
| Qwen2.5-Math-7B-SCAN-Base | 67.1 | 81.9 | 73.8 | 55.6 | 69.5 | 61.7 | 44.9 | 45.4 | 45.2 | 41.6 | 52.7 | **46.5** | 56.8 |
| Qwen2.5-Math-7B-SCAN-Pro | 72.9 | 90.7 | **80.9** | 58.6 | 73.6 | **65.3** | 44.2 | 47.8 | 45.9 | 37.8 | 53.1 | 44.2 | **59.1** |

reward score for the entire response. The response with the highest reward score is selected as the final answer. For the aggregation process, we take the lowest score among all steps as the overall reward score. Further experiments and discussions regarding different aggregation methods are provided in Appendix 4. The evaluation datasets cover various difficulty levels, including GSM8K [2] (elementary), MATH [11] (competition), College Math [31] (college), and Olympiad Bench [10] (Olympiad). We employ Qwen2.5-Math-7B-Instruct and Llama3.1-8B-Instruct as the policy model and set N to 8 (a reasonably practical setting in real-world applications). We report the majority voting [36] as the baseline and pass@8 as the upper bound.

**Step-wise Error Detection** We further investigate the PRM's ability to detect error locations within responses accurately. We use ProcessBench [44] as the evaluation benchmark, which measures the PRM's capability to identify the first error location in a given response. This evaluation focuses on the model's performance in recognizing fully correct samples and accurately identifying the error in incorrect responses. Formally, the evaluation metrics can be defined as: The final F1 score is the harmonic mean of the accuracies on correct and erroneous samples.

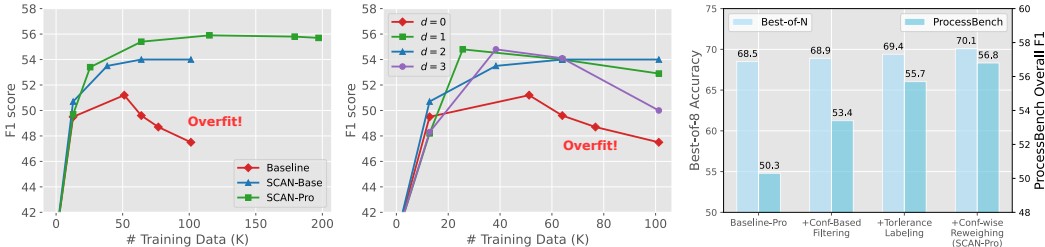

Figure 3: Ablation results in BoN evaluation and ProcessBench. For ProcessBench, we directly calculate the overall F1 score of full samples. **Left:** Scaling curve of the PRM training of different datasets. **Middle:** Scaling curve of selection of tolerance distance. **Right:** Effectiveness of each component. "Baseline" here represents the vanilla MC estimation method.

**Compared Baselines** Our primary comparisons are against 7B-scale process reward models, including Math-Shepherd [35], RLHFlow-PRM [38], Skywork-PRM [23], Math-PSA [34], and EurusPRM [40, 3]. We also include models trained with strong supervision, such as Qwen2.5-Math-PRM-7B[42] and UniversalPRM[30], as points of reference. However, we do not include direct comparisons with these strongly supervised models, as they rely on large-scale external critic models for guidance. As shown in Table 2, these critic models are already highly capable, and their supervision, typically via knowledge distillation, plays a key role in the final PRM performance. *However, we address a different issue from these approaches, i.e., denoising MC estimation itself.* For the Best-of-N evaluation, we directly test the publicly available checkpoints of the compared models. For ProcessBench, the results are directly sourced from Zheng et al. [44].

## 4.3 Main Results

**Best-of-8 Evaluation** Table 1 presents the Best-of-8 evaluation results based on the policy model Qwen2.5-Math-7B-Instruct. With only 101K synthetic samples from the SCAN-Base dataset, generated by the 1.5B model, the Qwen2.5-Math-7B-SCAN-Base model outperforms PRMs trained on larger synthetic datasets and approaches the performance of PRMs trained on human-annotated data (PRM800K). Notably, when trained with additional synthetic data generated by the 7B model, the resulting Qwen2.5-Math-7B-SCAN-Pro model surpasses PRM800K. This demonstrates that even with a 1.5B model, it is possible to synthesize data of comparable quality to human annotations.

**ProcessBench** Table 2 presents the evaluation results on ProcessBench. Both Qwen2.5-Math-7B-SCAN-Base and Qwen2.5-Math-7B-SCAN-Pro outperform all other process reward models, including those trained on PRM800K. Remarkably, their error detection capabilities even surpass those of the 70B-scale critic model Llama-3.3-70B-Instruct. Through the SCAN approach, Qwen2.5-Math-7B-Ins is able to generate process data for self-training, leading to a substantial improvement in its own error detection capability (from 19.9 to 59.1), demonstrating strong self-improvement potential.

## 4.4 Ablation Study

We conducted an ablation study to further demonstrate the effectiveness of each component. Additional results can be seen in Appendix B.

**Scaling Curve of SCAN-Base and SCAN-Pro** The left of figure 3 illustrates the performance variation of the model during training. We observe that without any denoising strategy, the model quickly overfits to noisy samples (see Baseline results). Our denoising strategy enables the model to grow steadily without overfitting to noisy samples and further improves the model's performance. By comparing the results of SCAN-Base and SCAN-Pro, we conclude that incorporating additional data sources enhances the diversity of the data, which in turn optimizes the model's upper bound.

**Choices of Tolerance Distance** Choosing an appropriate tolerance distance $d$ is critical, as both very small and very large values can introduce noise. When $d = 0$, it results in hard labeling, leading to severe noise, as shown in the scaling curve. Conversely, when $d = n$, it becomes soft labeling,

which also adds significant noise and hinders scaling [42]. The middle of figure 3 shows the results for different values of $d$. We find that $d = 2$ offers a good balance, reducing overfitting during training.

**Effectiveness of each component** The right of figure 3 shows the effectiveness of each component. We begin with a baseline where no denoising strategy is applied to the SCAN-Pro dataset. As we progressively incorporate denoising techniques, we observe consistent improvements in model performance across both best-of-n and ProcessBench evaluations. Both tolerance distance labeling and confidence reweighting contribute to performance gains. Tolerance distance labeling enhances the model's robustness when handling noisy samples, while confidence reweighting helps de-bias the probability estimation from different models on annotated samples.

## 5 Related Work

**Reward Models in Reasoning Tasks** Reward models play a crucial role in enhancing the capabilities of large language models (LLMs), particularly in complex reasoning tasks such as mathematical problem-solving [6, 39] and competitive programming [12]. In this context, reward models act as verifiers to assess the correctness of generated responses [4] or directly enhance LLM capabilities through preference alignment [19, 27]. Unlike outcome reward models (ORMs), which evaluate only the correctness of the final answer, process reward models [32, 20] (PRMs) provide more fine-grained evaluations by assessing each step of the reasoning process. Recent advancements [35, 18, 33] have demonstrated the significant potential of PRMs in scaling test-time compute [26, 42] and preference learning [35, 8], further highlighting their importance in the development of more capable and reliable LLMs.

**Process Reward Learning** Human-annotated data [20] is the primary solution, but due to the complexity of the task, manual annotation is highly costly. Recent works show the potential of Monte Carlo estimation as a promising alternative; however, it also introduces considerable noise [35, 18, 44]. This noise can be mitigated by incorporating critic models [21, 41, 7, 37], where LLMs are prompted to assess the correctness of each step directly. However, recent research [44] indicates that only large-scale models (e.g., Qwen2.5-Math-72B-Instruct) possess strong critic capabilities. Thus, this approach essentially distills the error-detection ability of large models into smaller process reward models through data distillation. Numerous works extend the strong supervision to more complex methods, like code verification [43], reverse verification [30], and data synthesis for preference learning [28]. Another line of work [40, 3] focuses on a different training paradigm by simply training an ORM on response-level labels, as optimized language models inherently function as reward models as well [24]. Our work takes a novel noisy-learning perspective and aims to address two key challenges in Monte Carlo annotation: high computational overhead and high noise ratio problem.

## 6 Conclusion

In this paper, we introduce an effective approach to improving process reward models from the perspective of data annotation. We first conduct a preliminary study to reveal the noise distribution in the Monte Carlo annotation process by introducing a confidence metric. We then propose an efficient data synthesis and robust learning framework to address the key challenges in Monte Carlo estimation. Through extensive experiments, we demonstrate the effectiveness of our proposed approach and the potential of model self-improvement from a robust learning perspective.

## Acknowledgement

We want to thank all the anonymous reviewers for their valuable comments. This work was supported by the National Science Foundation of China (NSFC No. 62206194), the Natural Science Foundation of Jiangsu Province, China (Grant No. BK20220488), and the Young Elite Scientists Sponsorship Program by CAST (2023QNRC001). We also acknowledge MetaStone Tech. Co. for providing us with the software, optimisation on high performance computing and computational resources required by this work.

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

# A Data Synthesis and Training Details

## A.1 Data Synthesis Details

Table 3: Data component of SCAN-Base and SCAN-Pro datasets.

| Dataset | Samples | Generator & Completer | Generate Response | | | Perform Rollouts | | | GPU hours |
|---------|---------|----------------------|------|-------|-------|---|-------|-------|-----------|
| | | | k | temp. | top_p | k | temp. | top_p | |
| SCAN-Base | 15K | Qwen2.5-Math-1.5B-Instruct | 64 | 0.7 | 0.8 | 8 | 0.7 | 0.8 | 174.4 |
| | 16K | Qwen2.5-Math-1.5B-Instruct | 64 | 1.0 | 1.0 | 8 | 0.7 | 0.8 | |
| | 31K | Qwen2.5-Math-1.5B-Instruct | 128 | 0.7 | 0.8 | 8 | 0.7 | 0.8 | |
| | 40K | Qwen2.5-Math-1.5B-Instruct | 128 | 1.0 | 1.0 | 8 | 0.7 | 0.8 | |
| SCAN-Pro (Increment) | 13K | Qwen2.5-Math-7B-Instruct | 64 | 0.7 | 0.8 | 8 | 0.7 | 0.8 | 200.1 |
| | 13K | Qwen2.5-Math-7B-Instruct | 64 | 1.0 | 1.0 | 8 | 0.7 | 0.8 | |
| | 71K | Llama3.2-3B-Instruct | 128 | 1.0 | 1.0 | 8 | 0.7 | 0.8 | |
| SCAN-Pro (Full) | | | | | | | | | 374.5 (Total) |

**Data Components and Inference Cost**   Table 3 presents the composition of the SCAN-Base and SCAN-Pro datasets, along with specific hyperparameter settings, including the total inference cost. Through our efficient framework design, we generated 197K samples using 374.5 GPU hours. All experiments were conducted on a single machine equipped with 8 GPUs, requiring only 47 hours in real time.

**Monte Carlo Tree Search**   Regarding the choice of inference strategy, we found that the commonly used Monte Carlo Tree Search (MCTS) method is inefficient for large-scale data synthesis. Annotating a single sample with MCTS involves multiple sequential steps, where each action depends on the state of the tree, making parallelization impractical. Additionally, maintaining extensive tree node information for each sample results in substantial memory overhead. As an alternative, we adopted the vanilla Monte Carlo method for annotation. This approach not only delivers strong annotation performance but also achieves results comparable to those of MCTS [18], while offering significantly higher efficiency.

**Implementation Details**   For inference deployment, we explored two approaches: (1) launching multiple vLLM servers and making API calls, using distributed routing strategies powered by FastChat [45], and (2) batching multiple queries and utilizing Ray to schedule resources for distributed inference. We found the second approach to be significantly more efficient, achieving over twice the speed of the first method. When estimating step-wise correctness via Monte Carlo (MC) estimation, we begin from the first step of the response and proceed sequentially until encountering the first position where $c_t = 0$, at which point the evaluation stops. While this introduces some dependency between steps, the depth of dependency is limited to the number of steps, which is considerably shallower and less complex compared to the recursive depth of Monte Carlo Tree Search (MCTS).

## A.2 PRM Training

We train our process reward models based on Qwen2.5-Math-7B-Instruct with a constant learning rate of $7 \times 10^{-6}$ and batch size of 128. The model is trained for one epoch, as we observed that multiple epochs lead to rapid overfitting, particularly on synthetic data. Although overfitting occurs more slowly on human-annotated data, it remains a concern with extended training.

# B Additional results

## B.1 Noise Distribution Across More Models

We present the noise distribution of additional models in Figure 4, including Llama-3.2-3B-instruct, Llama3.1-8B-Instruct, Qwen2.5-Math-1.5B-Instruct and Qwen2.5-Math-7B-Instruct. Overall, the observations discussed in section 2 remain consistent across these models as well.

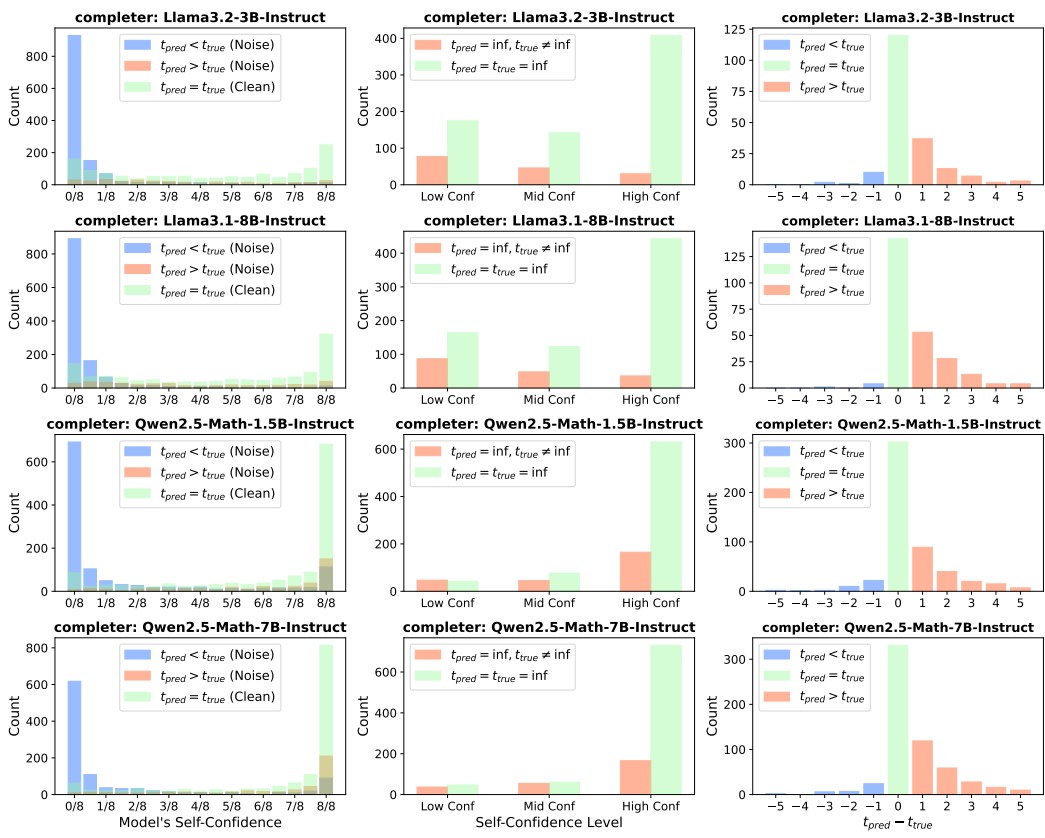

Figure 4: Noise distribution of additional models. Similar observations can be concluded across these models, further validating the consistency of our findings.

Table 4: Best-of-8 evaluation results of Qwen2.5-Math-7B-SCAN-Base and Qwen2.5-Math-7B-SCAN-Pro using different aggregation methods.

| Models | Aggregation Method | GSM8K | MATH | College Math | Olympiad Bench | Average |
|---|---|---|---|---|---|---|
| Qwen2.5-Math-7B-SCAN-Base | Min-Max | 97.1 | 86.9 | 47.8 | 44.4 | 69.1 |
| | Min-Vote | 96.9 | 87.5 | 47.7 | 44.9 | 69.3 |
| | Last-Max | 96.8 | 86.8 | 48.1 | 45.3 | 69.3 |
| | Last-Vote | 96.8 | 87.4 | 47.6 | 45.3 | 69.3 |
| Qwen2.5-Math-7B-SCAN-Pro | Min-Max | 97.1 | 87.3 | 48.1 | 47.7 | 70.1 |
| | Min-Vote | 97.2 | 87.8 | 47.9 | 46.7 | 70.2 |
| | Last-Max | 96.9 | 86.4 | 47.6 | 44.2 | 68.9 |
| | Last-Vote | 96.7 | 87.3 | 47.7 | 45.5 | 69.3 |

## B.2 Aggregation Methods

We experiment with the following four aggregation methods:

- **Min-Max:** This is the method used in our main experiments. The final selected response is given by $r^* = \text{argmax}_r \min(\mathbf{p})$, where $\mathbf{p} = [p_1, p_2, \ldots, p_t]$ represents the predicted step scores of the trained process reward model.
- **Last-Max:** The final selected response is $r^* = \text{argmax}_r p_t$, where $p_t$ is the reward score of the last step.

- **Min-Vote:** Inspired by the majority vote method, we designed a weighted voting strategy. The final answer is computed as $a^* = \operatorname{argmax}_a \sum_{i=1}^{|R|} \mathbb{I}(a_i = a) \times \min(\mathbf{p})$, where $\mathbb{I}(a_i = a)$ is an indicator function that equals 1 if the $i$-th response matches answer $a$ and 0 otherwise.
- **Last-Vote:** The final answer is computed as $a^* = \operatorname{argmax}_a \sum_{i=1}^{|R|} \mathbb{I}(a_i = a) \times p_t$.

Table 4 presents the results. We observe that the voting-based final answer yields better performance compared to the Min-based strategy. Therefore, if the primary objective is to obtain a single accurate answer, the Min-Vote strategy may be the more effective choice.

## B.3 BoN results on Llama3.1-8B-Instruct

Table 5: Best-of-8 evaluation results of the policy model Llama3.1-8B-Instruct.

| Model | Training Samples | Annotation Method | GSM8K | MATH | College Math | Olympiad Bench | Avg. |
|---|---|---|---|---|---|---|---|
| Greedy | | | 86.1 | 51.5 | 34.0 | 16.9 | 47.1 |
| Majority Vote@8 | | | 90.5 | 60.3 | 37.2 | 24.7 | 53.2 |
| **Pass@8 (Upper Bound)** | | | 95.7 | 75.6 | 48.3 | 40.2 | 65.0 |
| Qwen2.5-Math-PRM-7B | 1500K | MC, KD | 92.5 | 63.3 | 40.2 | 25.6 | 55.4 |
| UniversalPRM-7B | 690K | MC, KD | 93.3 | 65.6 | 40.4 | 26.5 | 56.5 |
| RLHFlow-PRM-Mistral-8B | 273K | MC-Only | 90.8 | 60.4 | 37.7 | 24.3 | 53.3 |
| RLHFlow-PRM-DeepSeek-8B | 253K | MC-Only | 90.6 | 60.6 | 37.1 | 24.7 | 53.2 |
| EurusPRM-Stage1 | 463K | Implicit | 93.0 | 64.7 | 40.7 | 28.3 | 56.7 |
| EurusPRM-Stage2 | 693K | Implicit | 93.4 | 66.4 | 41.3 | 28.6 | 57.4 |
| Math-PSA-7B | 1395K | MC-Only | 92.5 | 63.8 | 39.8 | 27.7 | 56.0 |
| Skywork-PRM-Qwen2.5-7B | - | - | 93.3 | 67.1 | 41.3 | 28.4 | 57.5 |
| Qwen2.5-Math-7B-PRM800K | 264K | Human | 92.0 | 64.0 | 40.9 | 28.2 | 56.3 |
| Qwen2.5-Math-7B-SCAN-Base | 101K | MC-Only | 93.1 | 64.8 | 40.9 | 27.6 | 56.6 |
| Qwen2.5-Math-7B-SCAN-Pro | 197K | MC-Only | 93.0 | 65.8 | 41.5 | 28.4 | 57.2 |

In addition to the Qwen model, we also evaluated the performance of our trained PRMs on Llama3.1-8B-Instruct. The responses were generated with a temperature setting of 0.5. Table 5 presents the results, showing that with only 197K data samples, our PRM achieves performance comparable to other PRMs and outperforms the human-annotated PRM800K dataset.

## B.4 Ablation in Data Components

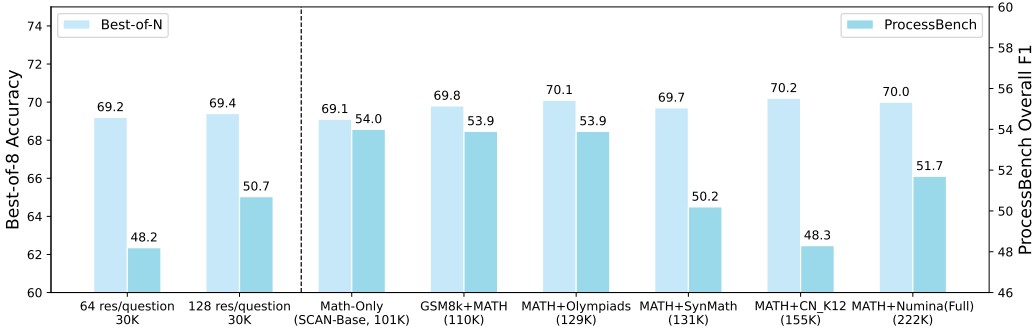

Figure 5: **Left:** Ablation results of different responses per question estimating self-confidence value. **Right:** Ablation results on external data sources.

**Accurate Estimation of Self-Confidence**  We estimate the model's self-confidence for a given question by sampling multiple responses. Our findings indicate that a more accurate estimation of self-confidence scores significantly improves data quality, primarily in: (1) **more precise selection**

**of high-confidence samples**, as high-confidence samples tend to have lower noise, and (2) **more accurate reweighting of step-wise correctness scores**, reducing bias in the learning process. We experiment with different numbers of responses per question, setting $k \in \{64, 128\}$. Notably, a larger $k$ leads to more precise self-confidence estimation. The left of figure 5 presents the results. To ensure a fair comparison, we maintain a fixed training dataset size of 30K samples. Our results demonstrate that a more accurate self-confidence estimation brings significant improvements in both Best-of-8 and ProcessBench evaluations.

**Incorporation More Data Sources**  We explore the potential of incorporating additional data sources, including GSM8K [2] and Numina-Math [16]. Numina-Math consists of various data sources, such as Olympiad problems and synthetic math datasets. We selected specific subsets from it for our experiments. Figure 5 presents the results. Incorporating GSM8K provides a certain degree of improvement; however, we find that generating responses for GSM8K incurs a higher computational cost compared to MATH. This is because the model makes fewer errors when generating responses for GSM8K, given its relatively simpler problem set. Moreover, we observe that incorporating subsets of synthetic math data significantly degrades performance. Upon further investigation, we suspect this is due to a high proportion of unsolvable problems or incorrect reference answers in the dataset. Overall, we find that **problem difficulty** and the **quality of question-answer pairs** are two key factors that significantly impact performance.

## B.5  Reduction in Noise Ratio

We further investigate whether our method effectively reduces the noisy data. To quantify this, we measure the noise ratio levels. While direct noise measurement is infeasible for our generated data due to the absence of ground truth, we employ ProcessBench as a proxy benchmark, given its similar data sources (including MATH) and response generation process (using Qwen and Llama series models). As shown in Table 6, our denoising process achieves a significant reduction in noisy data content.

Table 6: Noise ratio of synthetic data with and without SCAN denoising.

| Completer | Vanilla MC | + SCAN Denoising |
|---|---|---|
| Llama-3.1-8B-Ins | 56.2% | 19.1% (37.1%↓) |
| Qwen2.5-Math-7B-Ins | 51.8% | 29.4% (22.4%↓) |

## B.6  SCAN Inference Speed

Table 7 shows the comparison of performance and inference speed of SCAN Models and other large critic models. The inference speed is tested on 4 A100-40G GPUs. Experimental results demonstrate that discriminative models exhibit significantly superior inference speed compared to critic models. Particularly, long-chain-of-thought (long CoT) critic models such as DeepSeek-R1-Distill-Qwen-7B show even more pronounced efficiency bottlenecks.

Table 7: Performance and inference speed comparison between SCAN models and critic models.

| Model | Best-of-N (Avg. Acc.) | ProcessBench (Avg. F1) | Infer Speed |
|---|---|---|---|
| Qwen2.5-Math-7B-Instruct | - | 17.3 | 1.5 samples / s |
| Qwen2.5-7B-Instruct | - | 36.8 | 10.8 samples / s |
| DeepSeek-R1-Distill-Qwen-7B | - | 53.4 | 0.5 samples / s |
| Qwen2.5-Math-7B-SCAN-Base | 69.1 | 56.8 | 44 samples / s |
| Qwen2.5-Math-7B-SCAN-Pro | 70.1 | 59.1 | 44 samples / s |

### B.7 Incorporate SCAN with Knowledge Distillation Method

Zheng et al. [42] demonstrated the effectiveness of consensus filtering, where Monte Carlo estimation can be combined with knowledge distillation to further improve data quality. Since any improvement in MC estimation directly benefits downstream pipelines, we integrated our method (SCAN) into a consensus filtering framework to evaluate its effectiveness.

Table 8: Results of SCAN models with knowledge distillation (KD) from QwQ-32B.

| Model | Method | # Samples | Best-of-N (Avg. Acc) | ProcessBench (Avg. F1) |
|---|---|---|---|---|
| Qwen2.5-Math-PRM-7B | MC & KD | 1500K | 70.1 | 73.5 |
| Baseline-7B (w/o denoising) + KD | MC & KD | 100K | 69.0 | 52.5 |
| SCAN-Base-7B (w/ denoising) + KD | MC & KD | 100K | **70.3 (+1.3)** | **60.8 (+8.3)** |

As shown in Table 8, using SCAN's denoised MC data with KD (SCAN + KD) significantly outperforms using standard MC data with KD (Baseline + KD), especially on the fine-grained ProcessBench. Therefore, the improved version of MC can be readily used as a plug-in or substitute in any framework that involves MC estimation, offering further performance gains.

### B.8 SCAN Effectiveness in Broader Domains

We further examine the generalization ability of SCAN beyond mathematical reasoning. Specifically, we extend our Math PRM to a general-domain task, GPQA-Diamond [25], with results reported in Table 9. Our PRM consistently outperforms the majority-vote baseline, indicating that the reasoning capability it acquires is transferable beyond mathematics. This suggests that developing robust, domain-specialized PRMs represents an important direction for future research.

Table 9: Best-of-N results of SCAN-Pro in GPQA-Diamond.

| Method | N = 1 | N = 2 | N = 4 | N = 8 |
|---|---|---|---|---|
| Maj @ N | 33.8 | 33.8 | 38.9 | 37.3 |
| PRM @ N | 33.8 | **36.4** | **40.4** | **39.4** |

### B.9 Analysis of Process Error Types

To further understand the noise in MC-annotated data, we conducted a qualitative analysis by categorizing errors in 60 samples from ProcessBench, with results illustrated in Table 10.

Table 10: Error Type Analysis Results and PRM Accuracy.

| Error Type | Description | PRM Accuracy |
|---|---|---|
| Calculation Error | Mistakes in arithmetic or computation. | 15 / 20 |
| Logical Error | Inconsistencies or unjustified steps. | 9 / 20 |
| Conception Error | Misunderstanding of concepts or formulas. | 10 / 20 |

SCAN-PRM is highly effective at detecting calculation errors but less sensitive to abstract logical or conceptual mistakes, indicating that SCAN primarily captures procedural correctness. Achieving deeper semantic accuracy, however, remains more challenging and will require further investigation in future work.

## C Discussion

### C.1 Performance discrepancy of models in test-time scaling and ProcessBench

We observe a notable discrepancy between model performance in Best-of-N evaluation and Process-Bench. For example, models such as Qwen2.5-Math-PRM perform comparably to SCAN models

under Best-of-$N$, yet show larger performance gains on ProcessBench. This contrast reflects the fundamental difference between Monte Carlo (MC) and knowledge distillation (KD) annotation.

- **MC annotation is coarse-grained**: it judges correctness solely based on the final outcome of a solution. This provides a strong global signal of solution quality, which explains why MC-trained models excel in Best-of-N evaluation, where the objective is to identify the best overall response. **KD annotation is fine-grained**: a powerful critic model can analyze reasoning step by step, pinpointing the exact location of errors. Such supervision is crucial for ProcessBench, which explicitly evaluates step-level error detection.

These differences in supervision quality directly influence model performance. In Best-of-N evaluation (Table 1), MC-trained models perform well, as their coarse error modeling still captures useful global signals. However, in ProcessBench (Table 2), which requires precise step-level correctness, models trained only with MC supervision underperform due to their weaker error localization ability. Therefore, while our cost-efficient SCAN method achieves strong results in selecting the best overall solution, the more expensive fine-grained supervision used by Qwen-PRM naturally yields better step-level error detection. Notably, our experiments in Table 8 show that integrating SCAN with KD significantly reduces this performance gap.

## C.2 Effectiveness of SCAN Annotator Model

The effectiveness of SCAN is closely tied to the capability of the base annotator, and a central insight of our work is that this capability must be well-matched to the difficulty of the problem. To quantify this alignment, we propose a self-confidence metric.

In general, the quality of process annotation depends on two factors: (1) the capability of the base model and (2) the difficulty of the problem. Effective supervision arises when these two are properly aligned For instance, if a highly capable model is applied to an extremely simple problem, it may always recover the correct final answer, even when given erroneous prefix steps, as it can effortlessly detect and correct mistakes. Conversely, if the model is too weak for a challenging problem (e.g., Olympiad-level), it is unlikely to reach the correct answer under any prefix, producing noisy or uninformative annotations.

The key insight, therefore, is that annotation quality improves when the model's capability matches the task difficulty, and this match can be estimated using SCAN's self-confidence metric.

## D    Limitations and Future Work

Despite the promising results, our work still faces several limitations.

**Limitations of Monte Carlo Estimation**    During our experiments, we observed a type of noise that cannot be effectively handled by Monte Carlo (MC) estimation—false positives, where a response is predicted to be entirely correct but actually contains errors. This issue arises from a strong assumption underlying MC estimation: if the final answer is correct, all intermediate steps are assumed to be correct as well. Consequently, step-level errors embedded in seemingly correct responses remain undetected. This limitation makes purely MC-based annotation strategies insufficient for ensuring process-level fidelity.

**Limitations of Process Reward Models (PRMs) in AI alignment**    Although PRMs show encouraging improvements in test-time scaling and error localization, their effectiveness in AI alignment and safety remains limited. In particular, PRMs are still vulnerable to reward hacking, where a model exploits imperfections in the reward signal to achieve high scores without genuine reasoning. This issue becomes especially pronounced in out-of-distribution evaluations, where models may inadvertently optimize for misaligned objectives.

**A Promising Direction: Generative Process Reward Models (GenPRMs).**    To address both the annotation and alignment challenges, we view generative process-level reward models (GenPRMs) as a promising path forward. Unlike discriminative PRMs, GenPRMs are tasked with producing explicit rationales that identify both the location and nature of reasoning errors. This grounds the

reward signal in the actual reasoning process, thereby reducing the likelihood of reward hacking and improving interpretability. In data synthesis, large critic models can further enhance GenPRMs by pinpointing step-level mistakes, which can be integrated with MC sampling to yield higher-quality supervision. We believe GenPRMs represent a key direction for future research, offering stronger process-consistent reward signals and a safer, more interpretable framework for supervising high-capability reasoning models.

## E   Human Verification

We further analyze the capability of our trained PRMs through human evaluation. Specifically, we manually inspect whether the PRM accurately identifies correct and incorrect samples, as well as its ability to locate errors within responses precisely. Our evaluation focuses on two challenging settings in the best-of-n scenario:

- **High-confidence samples**, where almost every response is correct—can the PRM identify the few erroneous responses hidden among predominantly correct ones?
- **Low-confidence samples**, where only a few responses are correct—can the PRM successfully identify those correct responses?

We list some representative cases below, with human verification and analysis.

**Best-Of-N Evaluation Example 1**

Task: Best-of-8 Evaluation
Purpose: Whether PRM can select incorrect responses in high-conf questions.
Dataset Name: GSM8K
Response Generator: Qwen2.5-Math-7B-Instruct
Process Reward Model: Qwen2.5-Math-7B-SCAN-Pro
Problem:
Carla is downloading a 200 GB file. Normally she can download 2 GB/minute, but 40% of the way through the download, Windows forces a restart to install updates, which takes 20 minutes. Then Carla has to restart the download from the beginning. How load does it take to download the file?

---

Response (incorrect, 7 others correct):
To determine the total time it takes Carla to download the 200 GB file, we need to break down the process into segments and calculate the time for each segment.
Ground Truth: Correct, Predicted Score: 96.1, ✓

1. **Calculate the time to download 40% of the file:**
   - The file size is 200 GB.
   - 40% of 200 GB is $0.40 \times 200 = 80$ GB.
   - Carla downloads at a rate of 2 GB per minute.
   - Time to download 80 GB is $\frac{80}{2} = 40$ minutes.
Ground Truth: Correct, Predicted Score: 91.8, ✓

2. **Add the time taken for the restart:**
   - The restart takes 20 minutes.
   - Total time so far is $40 + 20 = 60$ minutes.
Ground Truth: Correct, Predicted Score: 94.1, ✓

3. **Calculate the time to download the remaining 160 GB:**
   - The remaining file size is $200 - 80 = 120$ GB.
   - Carla downloads at a rate of 2 GB per minute.
   - Time to download 120 GB is $\frac{120}{2} = 60$ minutes.
Ground Truth: Incorrect, Predicted Score: 3.4, ✓
Explanation: Since the reboot requires downloading from scratch, she needs to re-download the entire 200 GB instead of the remaining 120 GB.

4. **Add the time taken for the second restart:**
   - The restart takes 20 minutes.
   - Total time so far is $60 + 60 + 20 = 140$ minutes.

Therefore, the total time it takes Carla to download the file is $\boxed{140}$ minutes.

---

Analysis: PRM can select incorrect responses with accurate error location from numerous correct ones, demonstrating strong discriminative capability.

Task: Best-of-8 Evaluation
Purpose: Whether PRM can select correct responses in low-conf questions.
Dataset Name: Olympiad Bench
Response Generator: Qwen2.5-Math-7B-Instruct
Process Reward Model: Qwen2.5-Math-7B-SCAN-Base
Problem:
Find all triples $(a, b, c)$ of positive integers such that $a^3 + b^3 + c^3 = (abc)^2$.

---

Response (correct, 7 others incorrect):
To find all triples $(a, b, c)$ of positive integers such that $a^3 + b^3 + c^3 = (abc)^2$, we start by analyzing the equation and considering possible values for $a$, $b$, and $c$.
Ground Truth: Correct, Predicted Score: 95.3, ✓

First, let's check if there are any small solutions. We start with $a = b = c$:

$$a^3 + a^3 + a^3 = (a^3) = (a^3)^2 \implies 3a^3 = a^6 \implies a^6 - 3a^3 = 0 \implies a^3(a^3 - 3) = 0.$$

Since $a$ is a positive integer, $a^3 \neq 0$, so we have:

$$a^3 = 3 \implies a = \sqrt[3]{3}.$$

Since $\sqrt[3]{3}$ is not an integer, $a = b = c$ does not provide a solution.
Ground Truth: Correct, Predicted Score: 85.2, ✓

Next, let's try some small distinct values for $a$, $b$, and $c$. Suppose $a = 1$:
$$1^3 + b^3 + c^3 = (1 \cdot b \cdot c)^2 \implies 1 + b^3 + c^3 = (bc)^2.$$
Rearranging terms, we get:

$$b^3 + c^3 = (bc)^2 - 1.$$

We need to find positive integers $b$ and $c$ that satisfy this equation. Let's try $b = 2$:
$$2^3 + c^3 = (2c)^2 - 1 \implies 8 + c^3 = 4c^2 - 1 \implies c^3 - 4c^2 + 9 = 0.$$

We test possible integer values for $c$:
- If $c = 1$, then $1^3 - 4 \cdot 1^2 + 9 = 1 - 4 + 9 = 6 \neq 0$.
- If $c = 2$, then $2^3 - 4 \cdot 2^2 + 9 = 8 - 16 + 9 = 1 \neq 0$.
- If $c = 3$, then $3^3 - 4 \cdot 3^2 + 9 = 27 - 36 + 9 = 0$.
So, $c = 3$ is a solution. Therefore, $(a, b, c) = (1, 2, 3)$ is a solution. By symmetry, $(1, 3, 2)$, $(2, 1, 3)$, $(2, 3, 1)$, $(3, 1, 2)$, and $(3, 2, 1)$ are also solutions.
Ground Truth: Correct, Predicted Score: 67.2, ✓

(continue on the next page)

Now, let's check if there are any other solutions. Suppose $a = 2$:

$$2^3 + b^3 + c^3 = (2bc)^2 \implies 8 + b^3 + c^3 = 4b^2c^2.$$

Rearranging terms, we get:

$$b^3 + c^3 = 4b^2c^2 - 8.$$

We need to find positive integers $b$ and $c$ that satisfy this equation. Let's try $b = 1$:

$$1^3 + c^3 = 4 \cdot 1^2 \cdot c^2 - 8 \implies 1 + c^3 = 4c^2 - 8 \implies c^3 - 4c^2 + 9 = 0.$$

We already know that $c^3 - 4c^2 + 9 = 0$ has no positive integer solutions. Let's try $b = 2$:

$$2^3 + c^3 = 4 \cdot 2^2 \cdot c^2 - 8 \implies 8 + c^3 = 16c^2 - 8 \implies c^3 - 16c^2 + 16 = 0.$$

We test possible integer values for $c$:
- If $c = 1$, then $1^3 - 16 \cdot 1^2 + 16 = 1 - 16 + 16 = 1 \neq 0$.
- If $c = 2$, then $2^3 - 16 \cdot 2^2 + 16 = 8 - 64 + 16 = -40 \neq 0$.
- If $c = 4$, then $4^3 - 16 \cdot 4^2 + 16 = 64 - 256 + 16 = -176 \neq 0$.
Since $c^3 - 16c^2 + 16 = 0$ has no positive integer solutions for $c \geq 2$, there are no other solutions when $a = 2$.
Ground Truth: Correct, Predicted Score: 61.7, ✓

By symmetry, there are no other solutions when $a = 3$ or higher. Therefore, the only solutions are:

$$\boxed{(1, 2, 3), (1, 3, 2), (2, 1, 3), (2, 3, 1), (3, 1, 2), (3, 2, 1)}.$$

Ground Truth: Correct, Predicted Score: 81.3, ✓

---

Analysis: PRM can select incorrect responses from numerous incorrect ones, but it is less confident in some of the correct steps, as demonstrated by the relatively low step scores of intermediate steps.

