# OpenReview forum: "SCAN: Self-Denoising Monte Carlo Annotation for Robust Process Reward Learning"
_NeurIPS.cc/2025/Conference — NeurIPS 2025 poster_

### Official Review · Reviewer_MDHU · 2025-07-02

**Clarity:** 2
**Significance:** 3
**Originality:** 3
**Rating:** 5
**Confidence:** 3

**Summary:**

This paper introduces a method for training process reward models (PRMs) without requiring step-by-step human annotation data that builds upon prior self-supervised methods for training PRMs known as "Monte Carlo annotation / estimation", where rollouts from a completion model are used to estimate the correctness of a particular step, based on how often the generated completions result in the correct final answer.

In particular, the paper contributes a study as to when and why Monte Carlo annotations can become inaccurate or noisy, due to weaknesses in the completion model: When the completion model is weak at a certain question, it is likely to fail to reach the correct answer even when providing with a small number of correct steps, resulting in under-estimation of the first step index where reasoning fails. In contrast, when the completion model is able to self-correct after a true reasoning error, it ends up over-estimating the first step index where reasoning fails. Furthermore, by using another set of Monte-Carlo rollouts to estimate the completion model's confidence in the correct answer, the authors find that under-estimation usually occurs when the model has low self-confidence on a question, whereas correct estimation (and some over-estimation) generally occurs when the model has high self-confidence in a questions.

Using these findings and others as a guide, the authors develop a "self-denoising" scheme for PRM training, which (i) filters out low-confidence questions from the training data for the PRM; (ii) generates step-by-step annotations via Monte Carlo rollouts only for questions where the completion model gets the wrong answer, improving efficiency; (iii) corrects for over-estimation of the first failure index by adding soft labels; (iv) corrects for models that are too strong at self-correcting via confidence-reweighting. In experiments, they show that this method, SCAN, outcompetes most other PRMs in the 7-8B range on math reasoning and ProcessBench, including those trained on human-annotated data. They also find that SCAN on a 7B model is competitive with some (but not all) 70B models.

**Questions:**

- Could you address my concern about the justification of confidence-wise reweighting?
- Could you address the notational issues that affect the clarity of the paper?
- How is 100% sample utilization achieved?
- Could you more carefully explain which negative samples are collected for Monte Carlo step-wise annotation?

There are also a number of typos, e.g. "Noise-torlerant" should be "noise-tolerant", and "noisy positive" seems like a non-standard term compared to "false positive".

**Ethical Concerns:**

["NO or VERY MINOR ethics concerns only"]

**Final Justification:**

The authors have addressed all of my concerns and questions, and I continue to recommend acceptance.

**Limitations:**

I think the authors should include a Broader Impacts statement about the implications of this PRM training strategy on AI alignment and safety. PRMs were initially introduced partly to overcome the limitations of outcome-based RMs, which can lead to safety issues due to over-optimization and reward-hacking, as we've already seen in recent studies of reasoning models (Denison et al, 2024; Bondarenko et al 2025).
Because PRMs are supposed to reward only valid chains of thought, the hope is that they are more resistant to this reward hacking behavior. However, Monte Carlo annotation loses this a lot of this benefit of PRMs, since the process-level labels are effectively generated entirely from outcome-based self-supervision, and also creates noise/errors in the reward specification that a policy can learn to exploit. This could also incentivize unsafe specification-gaming / reward hacking in reasoning models (e.g. rewriting test cases, so that the test is eventually passed). I think some discussion of these safety implications is warranted, since this paper uses the Monte Carlo annotation approach.

**Quality:**

3

**Strengths And Weaknesses:**

I don't follow the literature on LLM reward modeling closely, so I'm not well positioned to evaluate originality and significance. Nonetheless, I found the preliminary study quite insightful, and helpful for understanding the failure modes of outcome-based self-supervision for generating process-level labels. Given that process-based supervision has sometimes been pitched as a "safer" alternative to outcome-based supervision (which can incentivize reward hacking, etc.), it's valuable to understand out how process-based supervision can also fail or lead to a bad / misaligned training signal, especially when using synthetically generated annotations.

As for the SCAN method itself, empirically it appears to perform as well as training on human process-level annotations, and I found the experiments and ablations to be thorough.

I do, however, think the justification of the confidence-wise reweighting requires some work. The authors say that, " $c_i*$ represents the true correctness probability, while $c_i$ is the estimated probability derived from the completer model $π$ ", implying that $c_i*$ is some kind of model-independent property. However, it seems to me that the "probability of correctness" is never model-independent --- it's always relative to some generative process of how the remainder of the chain of thought is completed, whether that generative process is parameterized by a neural network, or by how a population of human reasoners would tend to complete a chain-of-thought. So what exactly is this "true correctness probability"? I don't think this is a coherent concept unless one defines some kind of "ground-truth" or "reference" generative process $\pi^*$. Furthermore, if correctness is defined in logical terms (e.g. whether a reasoning step is mathematically valid), then a particular step is either correct on incorrect --- so the concept of "probability of correctness" doesn't really make sense in that context either. Instead, in the context of Monte Carlo annotation, $c_i$ is actually *the probability that some model (or human) eventually reaches a correct answer* --- which is at best a *proxy* of whether that step is actually correct.

One way to fix the issue above is to make clear that there is some expert / reference policy that we care about, $\pi*$, (e.g. the distribution of how an expert human would generate reasoning traces). If that expert policy were used to generate Monte Carlo annotations of the dataset, we could estimate $c_i*$ as the probability that the expert would eventually generate a correct answer. We would like to use $c_i*$ as a soft label for training that is independent of the completion model $\pi$ (even though it's not independent of $\pi*$), but we only have $c_i$ for some other completion model $\pi$ -- which might overestimate $c_i*$ when the completion model is stronger than the expert, or underestimate $c_i*$ when the completion model is weaker than the expert. So to estimate $c_i*$, you divide by the completion model's self-confidence. IMO that would be a stronger justification for the confidence-wise reweighting.

There are a number of other clarity issues that could be fixed. For example, it was confusing that the symbol $y_i$ is used to refer the (hard) label for whether step $i$ is correct, but then later $y_{pred}$ and $y_{true}$ are used to denote the index/step of the first error. I would suggest change it to $t_e^{pred}$ and $t_e^{true}$, to denote that this an index-valued variable. and to make it consistent with the $t_e$ notation used in Equation 5.

I also didn't understand the 100% sample utilization point -- is it the case that all samples that are either negative or not high in self-confidence undergo Monte Carlo annotation? If so, that should be made clear. The line "we collect the negative samples with $y_{pred} = inf$ from these responses for further step-wise annotation" was also confusing to me. Are all negative samples collected, or only those with  $y_{pred} = inf$? And are these so-called "noisy positive samples", since  $y_{pred} = inf$ but $y_{true} < inf$? If so, why are they suddenly called "negative samples"?

Overall though, I don't see reasons to reject this paper -- I just think the clarity and explanation / justification could be improved at various points.

---

> ### Author Rebuttal · Authors · 2025-07-30
>
> Thank you so much for your exceptionally thorough review and insightful, constructive feedback. Your suggestions have been invaluable for improving the paper's clarity and rigor.
>
> **Concern 1 & Question 1:**
> about the justification of confidence-wise reweighting
>
> **Response to Concern 1:**
> We sincerely apologize for the mistakes in the definition of $c*$ and the lack of clarity in our original explanation.
> In our formulation, $c*$ denotes the predicted step correctness score provided by an ideal or golden annotator. Based on our previous findings, we observed that the noise in step-level annotations primarily stems from the mismatch in model annotation capabilities — leading to systematic under-estimation or over-estimation errors.
>
> We assume the existence of a golden MC annotator $\pi_{\text{gold}}$ that can perfectly model the correctness of each intermediate step via MC estimation, without introducing any systematic bias (i.e., under-estimation or over-estimation). The correct formulation of $c^*$ should be:
>
> $$c_t^* = P(y_{i} = \text{correct}| \pi_{\text{gold}}, q, x_{\leq t})$$
>
> Here we retain soft labels within a predefined tolerance distance, assuming these steps can be partially correct.
>
> However, we can only estimate this probability using our imperfect completer model $\pi_{\theta}$, which yields an estimated probability $\hat{c_t} = P(y_{i} = \text{correct} | \pi_{\theta}, q, x_{\leq t})$. This estimation is biased by $\pi_{\theta}$'s own capabilities (i.e., under-estimation or over-estimation). To address this, we introduce confidence-wise reweighting as a proxy to (1) reduce the bias introduced by imperfect base models, and to (2) normalize the annotation quality across different policy models.
>
> This mechanism allows us to mitigate annotation noise while preserving the learning signal from partially correct responses, ultimately leading to more robust training of process reward models.
>
> ---
>
> **Concern 2 & Question 2:**
> notational issues that affect the clarity of the paper?
>
> **Response to Concern 2:**
> We will change $y_{pred}, y_{true}$ to $t_e^{pred}, t_e^{true}$ in the relevant equation and figure.
> Thank you for pointing this out — we believe this change will make the presentation clearer and more precise.
>
> ---
>
> **Concern 3 & Question 3:**
> How is 100% sample utilization achieved?
>
> **Response to Concern 3:**
> Yes, we only collect negative samples (i.e., those where the predicted answer does not match the ground truth), indicating that process-level errors are definitely present.
>
> We achieve this by being more efficient. Unlike prior work (e.g., Math-Shepherd) that performs expensive MC annotation on all generated samples (even correct ones), we only perform MC annotation on samples that result in a wrong final answer. We assume that if a model gets an answer right, its reasoning is likely sound by its own standards, so re-annotating with the same model is redundant. Thus, every sample we choose to annotate via MC is used for training, achieving 100% utilization of the annotated data and saving significant computation.
>
> [1] Math-Shepherd: Verify and Reinforce LLMs Step-by-step without Human Annotations
>
>
> ---
>
> **Concern 4 & Question 4:**
> Explain which negative samples are collected for Monte Carlo step-wise annotation?
>
> **Response to Concern 4:**
> We apologize for the confusion. The statement "the negative samples with $y_{pred} = \text{inf}$" is incorrect — it should be $y_{pred} \neq \text{inf}$.
>
> The negative samples refer to all responses where $y_{pred} \neq \text{inf}$, indicating that the annotater model detect some error in the response. Our intention is to clearly convey the definition of negative samples (i.e., $y_{pred} \neq \text{inf}$), but we mistakenly used the wrong symbol.
> We sincerely apologize for the misstatement and any confusion it have caused.
>
> ---
>
> Thank you for your suggestion, which made us aware of some issues in our phrasing and the lack of sufficient clarification.
> We have carefully reviewed the entire paper and revised several potentially misleading expressions to ensure clearer and accurate understanding of our method.
>
> ---
>
> **Response to Concern about Limitations**
>
> This is an excellent and important point. We agree completely and will add a dedicated section discussing these implications. You rightly note that because MC annotation derives process labels from outcome supervision, it is still vulnerable to reward hacking, where a model finds flawed reasoning paths that happen to produce correct final answers. This could incentivize specification-gaming and undermines the safety promises of process-based supervision.
>
> **Limitations of PRMs in AI alignment:**
>
> While PRMs have shown promising improvements in test-time scaling and error localization, their effectiveness in AI alignment and safety remains limited, primary due to the reward hacking [1].
>
> Reward hacking occurs when a model exploits imperfections or ambiguities in the reward signal to achieve high scores without performing the intended reasoning or task. This issue is not unique to ORMs; it also affects PRMs, especially when the model generalizes well but inadvertently optimizes for a misaligned objective. This problem becomes particularly pronounced when evaluated on out-of-distribution data.
>
> **Limitations of MC Method in Process Annotation:**
>
> As the reviewer rightly pointed out, the current process-level annotations are ultimately derived from outcome correctness. Consequently, it is difficult to identify step-level errors in responses that yield correct final answers but contain faulty intermediate reasoning. This limitation makes purely Monte Carlo-based annotation strategies insufficient for ensuring alignment fidelity.
>
> **A Promising Direction: Generative Process Reward Models (GenPRMs)**
>
> To address both the alignment and annotation challenges, we believe generative process-level reward models (GenPRMs) offer a promising path forward:
> - For AI Alignment: Unlike discriminative PRMs, GenPRMs are tasked with producing explicit rationales that identify both the location and nature of reasoning errors. This enhances interpretability and grounds the reward signal in the actual reasoning process, making it harder for models to exploit superficial correlations—thereby reducing reward hacking and improving safety.
> - In Process Data Synthesis: GenPRMs provide more accurate error localization. In data generation settings, large critic models can be used to pinpoint precise step-level mistakes, which can then be integrated with Monte Carlo sampling to synthesize higher-quality process supervision. We discuss this further in Appendix D.
>
> In fact, we consider GenRMs [2] (or GenPRMs) a key direction for future research. We believe they offer strong potential to mitigate reward hacking by enforcing process-consistent reward signals and offering a more interpretable framework for reasoning supervision—both of which are crucial for the safe deployment of high-capability reasoning models.
>
> [1] DeepSeek-R1: Incentivizing Reasoning Capability in LLMs via Reinforcement Learning
>
> [2] Inference-time scaling for generalist reward modeling

---

> > ### Comment · Reviewer_MDHU · 2025-08-03
> >
> > Thank you for your detailed and thorough response to my concerns and clarificatory questions. I believe all the concerns I had are adequately addressed, and agree that the proposed revisions should be made in the updated version of the paper. I will maintain my current rating and recommend acceptance.

---

### Official Review · Reviewer_5Gg8 · 2025-07-05

**Clarity:** 3
**Significance:** 3
**Originality:** 3
**Rating:** 5
**Confidence:** 3

**Summary:**

This paper presents SCAN, a framework for robust process reward learning that analyzes and mitigates noise in Monte Carlo (MC) estimation. The core contributions are an efficient data synthesis strategy and a noise-tolerant training objective.

**Questions:**

The baseline categorization in Table 2 is somewhat unconventional, as it appears to group models with different training paradigms (e.g., some baselines are not strictly MC-based), which may complicate direct comparisons.

Could the authors explain the performance discrepancy of in Table1 and Table2? For example, Qwen-PRM performs on par with SCAN models in the Best-of-N evaluation (Table 1) yet wins by a large margin on ProcessBench (Table 2).


Suggestions
The discussion on related work could be enhanced by including recent studies on alternative PRM evaluation paradigms, such as:
[1] R-PRM: Reasoning-Driven Process Reward Modeling
[2] Inference-Time Scaling for Generalist Reward Modeling

**Ethical Concerns:**

["NO or VERY MINOR ethics concerns only"]

**Final Justification:**

The author address my concerns

**Limitations:**

yes

**Quality:**

3

**Strengths And Weaknesses:**

**Strengths**

The paper's primary strength lies in its insightful analysis of the noise distribution in MC-generated data, offering a novel and interesting perspective on the problem.

** Weaknesses**

A notable performance gap remains between this MC-only approach and methods that leverage Knowledge Distillation (KD) or human annotations.

---

> ### Author Rebuttal · Authors · 2025-07-30
>
> Thank you for your feedback and the opportunity to clarify our contributions and positioning.
>
> **Concern 1:**
> A notable performance gap remains between this MC-only approach and methods that leverage Knowledge Distillation (KD) or human annotations.
>
> **Response to Concern 1:**
> This is a crucial point. We offer three clarifications:
>
> 1. **SCAN is Complementary to KD, Not Competing**: Monte Carlo (MC) estimation and Knowledge Distillation (KD) are not mutually exclusive. In fact, many state-of-the-art methods like Qwen-PRM [1] and GenPRM [2] use MC as part of a consensus filter alongside KD. The goal of our paper is to improve the MC component itself. **A better MC method can directly benefit any pipeline that uses it**. To demonstrate this, we ran a new experiment integrating SCAN into a KD pipeline.
> To further verify the effectiveness of our MC-based approach, we integrated our method into a consensus filtering pipeline. The results are shown below:
>
> | Model | Method | Samples | BoN (Avg. Acc) | ProcessBench (Avg. F1) |
> | ----------- | ----------- | ----------- | ----------- | ----------- |
> | Qwen-PRM | MC & KD | 1500K | 70.1 | 73.5 |
> | Baseline (w/o denoising) + KD | MC & KD | ~100K | 69.0 | 52.5 |
> | SCAN (base, w/ denoising) + KD | MC & KD | ~100K | **70.3 (+1.3)** | **60.8 (+8.3)** |
>
> As shown, using SCAN's denoised MC data with KD (**SCAN + KD**) significantly outperforms using standard MC data with KD (**Baseline + KD**), especially on the fine-grained ProcessBench. This confirms that **our improvements to MC are additive and boost KD-based methods**.
>
> 2. **Vast Difference in Cost:** It is important to compare performance in light of computational cost. Qwen-PRM uses 1.5M samples annotated by a 72B critic model. Our largest model uses ~200K samples annotated by a 7B model. We estimate the data generation cost for Qwen-PRM is **40-50x higher** than ours. SCAN achieves competitive BoN performance and strong ProcessBench scores at a fraction of the cost.
> 3. **Research Focus:** Our primary goal is to investigate the fundamentals of MC annotation—a widely used but under-explored technique. We show that by understanding and mitigating noise *within* the MC process, we can substantially improve its effectiveness without relying on expensive external supervisors.
>
> The results of SCAN demonstrate that our denoising strategy can significantly improve the upper bound of MC-based methods. In other words, **the improved version of MC can be readily used as a plug-in or substitute in any framework that involves MC estimation, offering further performance gains.**
>
> In conclusion, SCAN **(1) significantly pushes the performance ceiling of low-cost, MC-only methods** and **(2) provides a better, drop-in replacement for the MC component in more complex KD-based pipelines**.
>
> [1] The Lessons of Developing Process Reward Models in Mathematical Reasoning
>
> [2] GenPRM: Scaling Test-Time Compute of Process Reward Models via Generative Reasoning
>
> ---
>
> **Question 1:**
> The baseline categorization in Table 2 is somewhat unconventional, as it appears to group models with different training paradigms (e.g., some baselines are not strictly MC-based), which may complicate direct comparisons.
>
> **Response to Question 1:**
> Thank you for pointing this out. We sincerely apologize for the oversight — some baselines that are not strictly trained using MC estimation are mistakenly placed in the "MC-only" category, which may have caused confusion.
> We will correct this to ensure a clearer and more accurate comparison across different training paradigms in the revised version.
>
> ---
>
> **Question 2:**
> Could the authors explain the performance discrepancy of in Table1 and Table2? For example, Qwen-PRM performs on par with SCAN models in the Best-of-N evaluation (Table 1) yet wins by a large margin on ProcessBench (Table 2).
>
> **Response to Question 2:**
> This discrepancy highlights the core difference between MC and KD annotation.
>
> - **MC annotation is coarse-grained.** It determines correctness by checking the final outcome of a completed solution. This provides a strong signal for the *overall quality* of a solution trace, which is why MC-trained models perform well on Best-of-N evaluation, where the goal is to pick the best overall response.
> - **KD annotation is fine-grained.** A powerful critic model can analyze the logic of each step, allowing it to pinpoint the *exact location* of an error. This supervision is crucial for ProcessBench, which evaluates step-level error detection accuracy.
>
> Such differences in supervision quality ultimately affect the performance of the trained models.
> - In the Best-of-N evaluation (Table 1), where models are evaluated based on the overall reward score aggregated from multiple responses, MC-trained models perform reasonably well because their coarse error modeling still captures useful global signals.
> - However, in ProcessBench (Table 2), which requires fine-grained, step-level correctness evaluation, models trained with MC supervision alone fall short due to their weaker error localization capability — hence the observed performance gap.
>
> Therefore, while our cost-efficient SCAN method produces models that are excellent at identifying the best overall solution (BoN), the expensive, fine-grained supervision used by Qwen-PRM naturally leads to better performance on precise error localization (ProcessBench). Our new experiments in W1 show that when SCAN is combined with KD, this gap closes significantly.
>
> ---
>
> **Question 3:**
> The discussion on related work could be enhanced by including recent studies on alternative PRM evaluation paradigms, such as: [1] R-PRM: Reasoning-Driven Process Reward Modeling [2] Inference-Time Scaling for Generalist Reward Modeling
>
> **Response to Question 3:**
> Thank you for the excellent suggestion. These are important related works. We will add a detailed discussion and the following results to the appendix. A key takeaway is that while generative PRMs are a promising direction, our discriminative PRM is far more inference-efficient.
>
> Generative PRM (4 x A100-40G):
>
> | Model      | ProcessBench (Avg. F1) | Infer Speed |
> | ----------- | ----------- | ----------- |
> | Qwen2.5-Math-7B-Instruct | 17.3 | 1.5 samples / s |
> | Qwen2.5-7B-Instruct | 36.8 | 10.8 samples / s |
> | DeepSeek-R1-Distill-Qwen-7B | 53.4 | 0.5 samples / s |
> | GenPRM-7B | 75.2 | 1.2 samples / s |
> | R-PRM-7B | 70.4 | 0.8 samples / s |
> | **Qwen2.5-Math-7B-SCAN-Base** | **56.8** | **44 samples / s** |
> | **Qwen2.5-Math-7B-SCAN-Pro** | **59.1** | **44 samples / s** |
>
> This highlights a critical trade-off: SCAN offers highly practical and scalable inference, while current generative PRMs trade speed for higher accuracy.
>
> [1] R-PRM: Reasoning-Driven Process Reward Modeling
>
> [2] GenPRM: Scaling Test-Time Compute of Process Reward Models via Generative Reasoning

---

> > ### Comment · Reviewer_5Gg8 · 2025-08-04
> >
> > Thank you for your response, i have change my rating

---

### Official Review · Reviewer_zmmN · 2025-07-05

**Clarity:** 3
**Significance:** 3
**Originality:** 3
**Rating:** 5
**Confidence:** 4

**Summary:**

The research paper introduces a novel framework, SCAN, to address the challenges of training high-quality Process Reward Models (PRMs) from synthetically generated data. The authors identify that standard Monte Carlo (MC) estimation methods produce noisy labels, which causes models to overfit and limits their performance. The paper first presents a preliminary study that analyzes and categorizes the noise in MC-generated data into two types: under-estimation (incorrectly flagging correct steps as errors) and over-estimation (failing to detect the true first error). To quantify this, they propose a "self-confidence" metric, which measures the annotation model's own success rate on a given problem.

**Questions:**

1. As mentioned in weaknesses, how well does the SCAN framework transfer to tasks outside of mathematical reasoning (e.g., coding, multi-modal reasoning)?
2. How does the choice of base annotation model (size/capability) affect the ultimate PRM performance? Is there a “minimum” effective annotator?

**Ethical Concerns:**

["NO or VERY MINOR ethics concerns only"]

**Limitations:**

Yes

**Quality:**

3

**Strengths And Weaknesses:**

## Strengths
1. It proposes an efficient and practical solution (SCAN) that improves both data quality and training robustness via self-denoising and confidence-based approaches. It tackles a real, practical challenge—reducing the cost and scaling limitations of human-annotated PRM data.
2. This paper presents strong empirical results and made code and data available, aiding reproducibility. It achieves comparable results with only ~6% of the inference cost compared to traditional MC methods. It outperforms or matches models trained on much larger or human-annotated datasets, on challenging benchmarks like ProcessBench and Best-of-N evaluation. In addition, it shows continued improvement with increased data, without apparent overfitting.

## Weaknesses
1. The self-denoising approach’s effectiveness is inherently limited by the capability of the base model used for annotation. Additional empirical exploration of this limit could strengthen the claims.
2. While SCAN-trained PRMs surpass baselines, a deeper discussion or ablation on the qualitative differences (e.g., error types, robustness to distribution shift) between human- and synthetically-annotated data could be valuable.
3. The method is demonstrated mainly for mathematical reasoning datasets. It is unclear how well SCAN generalizes to other domains or types of reasoning tasks.

---

> ### Author Rebuttal · Authors · 2025-07-30
>
> Thank you for the positive and constructive review, highlighting SCAN's practicality, efficiency, and strong empirical results.
>
> **Concern 1:**
> The self-denoising approach's effectiveness is inherently limited by the capability of the base model used for annotation. Additional empirical exploration of this limit could strengthen the claims.
>
> **Responses to Concern 1:**
> This is an excellent point. The effectiveness of SCAN is indeed tied to the base annotator's capability, and a key insight of our work is that this capability should be well-matched to the problem's difficulty. Our proposed self-confidence metric is designed precisely to measure this match.
>
> In general, the quality of process annotation depends on two key factors: (1) the capability of the base model, and (2) the difficulty of the problem. Effective annotation tends to occur when these two factors are well-matched — a core idea behind our SCAN framework. For example, if a very strong model is used on an extremely simple problem, it may always recover the correct final answer, even when provided with erroneous prefix steps, because it can easily detect and correct the errors. On the other hand, if the model is not powerful enough to solve a challenging problem (e.g., one at Olympiad level), it is unlikely to produce the correct answer regardless of the input, resulting in noisy or uninformative supervision.
>
> So the key insight is that the model's capability should ideally match the difficulty of the problem, and this alignment can be estimated via self-confidence metirc proposed in SCAN. Given a specific problem, we can try several base models of varying capabilities and select the one whose self-confidence score falls within an appropriate range (e.g., [0.75, 0.9]).
>
> In SCAN, we retain samples where self-confidence is 1 to maximize data utilization. To mitigate the risk of over-estimation noise (where the model self-corrects and misses the true error), we introduce a tolerance distance for soft labels. This provides a robust learning signal even from imperfect annotations.
>
> In summary, SCAN provides a principled framework to manage the annotator's capability by: **(1) using self-confidence to select an appropriately-sized annotator for the task**, and **(2) using a tolerance distance to robustly learn from its outputs**.
>
> ---
>
> **Concern 2:**
> While SCAN-trained PRMs surpass baselines, a deeper discussion or ablation on the qualitative differences (e.g., error types, robustness to distribution shift) between human- and synthetically-annotated data could be valuable.
>
> **Response to Concern 2:**
> Thank you for this valuable suggestion. To better understand our PRM's capabilities, we conducted a qualitative analysis by categorizing errors in 30 samples from ProcessBench.
>
> | Error Type | Description | PRM Accuracy |
> | ----------- | ----------- | ----------- |
> | Calculation Error | Mistakes in arithmetic or computation. | 9 / 10 |
> | Logical Error | Inconsistencies or unjustified steps. | 5 / 10 |
> | Conception Error | Misunderstanding of concepts or formulas. | 6 / 10 |
>
> Our PRM is highly effective at detecting calculation errors but is less sensitive to more abstract logical and conceptual mistakes. This suggests that while SCAN effectively captures procedural correctness, future work could focus on improving the detection of deeper semantic errors. We will expand this analysis and add case studies to the final paper.
>
> ---
>
> **Concern 3:** The method is demonstrated mainly for mathematical reasoning datasets. It is unclear how well SCAN generalizes to other domains or types of reasoning tasks.
>
> **Question 1:** As mentioned in weaknesses, how well does the SCAN framework transfer to tasks outside of mathematical reasoning (e.g., coding, multi-modal reasoning)?
>
> **Response to Concern 3 & Question 1:**
> We appreciate the reviewer’s interest in evaluating the generalization capability of SCAN beyond the mathematical reasoning domain. To this end, we extend our Math PRM to a general-domain task, GPQA-Diamond, and report the following results:
>
> |  | N=1 | N=2 | N=4 | N=8 |
> | ----------- | ----------- | ----------- | ----------- | ----------- |
> | Majority Vote @ N | 33.8 | 33.8 | 38.9 | 37.3
> | PRM @ N | 33.8 | **36.4** | **40.4** | **39.4** |
>
> Our PRM consistently outperforms the majority vote baseline, demonstrating that the reasoning capabilities it learned are transferable beyond mathematics.
>
> That said, building high-quality, domain-specific PRMs for tasks like coding or multi-modal reasoning remains challenging, primarily due to:
>
> - **Data Scarcity:** High-quality process-level data is rare in these domains. As noted in a concurrent work [1], math-trained PRMs can outperform PRMs trained on synthetic code data, likely due to the quality gap (e.g., PRM800K vs. MC-generated code data).
> - **Evaluation Reliability:** Annotating and evaluating processes in other domains can be unreliable. For example, multi-modal models often fail to follow specific output formatting instructions, complicating automated evaluation.
>
> While we believe SCAN's data synthesis strategy can be extended to these domains, we see the development of robust, domain-specific PRMs as a significant future research direction. We will add this discussion to the paper.
>
> [1] From Mathematical Reasoning to Code: Generalization of Process Reward Models in Test-Time Scaling.
>
> ---
>
> **Question 2:**
> How does the choice of base annotation model (size/capability) affect the ultimate PRM performance? Is there a "minimum" effective annotator?
>
> **Response to Question 2:**
> Thank you for the thoughtful question. As discussed in our Response to Concern 1, we explain how to select an appropriate annotator to ensure the effectiveness of the self-denoising approach.
> The effectiveness of a base annotation model should be considered in conjunction with the difficulty of the target questions. For example, in relatively easier benchmarks such as GSM8K or Math, a 1.5B-sized model may already serve as a sufficient annotator, as it can achieve reasonably high accuracy. In contrast, for more challenging problems such as AIME24 or AIME25, stronger annotators (e.g., QwQ-32B) may be required to produce reliable supervision.

---

### Decision · Program_Chairs · 2025-09-17

**Decision:**

Accept (poster)

**Comment:**

This paper presents SCAN, a framework for robust process reward learning that analyzes and mitigates noise in Monte Carlo (MC) estimation. The core contributions are an efficient data synthesis strategy and a noise-tolerant training objective.

All the reviewers thought of the paper highly and felt that it should be accepted. Please incorporate the discussion into the final version of the paper.